# Comparative Effects of Particle Sizes of Cobalt Nanoparticles to Nine Biological Activities

**DOI:** 10.3390/ijms21186767

**Published:** 2020-09-15

**Authors:** In Chul Kong, Kyung-Seok Ko, Dong-Chan Koh, Chul-Min Chon

**Affiliations:** 1Department of Environmental Engineering, Yeungnam University, Gyungsan 38541, Korea; ickong@ynu.ac.kr; 2Geologic Environment Division, Korea Institute of Geoscience & Mineral Resources (KIGAM), Daejeon 34132, Korea; chankoh@kigam.re.kr (D.-C.K.); femini@kigam.re.kr (C.-M.C.)

**Keywords:** biological activity, cobalt metal oxide, nanoparticle, particle size

## Abstract

The differences in the toxicity of cobalt oxide nanoparticles (Co-NPs) of two different sizes were evaluated in the contexts of the activities of bacterial bioluminescence, *xyl*-*lux* gene, enzyme function and biosynthesis of β-galactosidase, bacterial gene mutation, algal growth, and plant seed germination and root/shoot growth. Each size of Co-NP exhibited a different level of toxicity (sensitivity) in each biological activity. No revertant mutagenic ratio (greater than 2.0) of *Salmonella typhimurium* TA 98 was observed under the test conditions in the case of gene-mutation experiments. Overall, the inhibitory effects on all five bacterial bioassays were greater than those on algal growth, seed germination, and root growth. However, in all cases, the small Co-NPs showed statistically greater (total average about two times) toxicity than the large Co-NPs, except in shoot growth, which showed no observable inhibition. These findings demonstrate that particle size may be an important physical factor determining the fate of Co-NPs in the environment. Moreover, combinations of results based on various biological activities and physicochemical properties, rather than only a single activity and property, would better facilitate accurate assessment of NPs’ toxicity in ecosystems.

## 1. Introduction

Many nanotechnology-based products are available in various areas of the industrial market and their increasing use in commercial products has led to significant exposure to nanoparticles (NPs) in the environment. Among the four categories of NPs (carbon-based, metal-based, dendrimers, and composites), metal-based NPs (MNPs; metals, metal oxides, and other metal-containing NPs) are commercially available in the fields of industrial and biomedical technologies and have been studied widely [1,2,3,4]. Among these MNPs, cobalt-based MNPs are currently attracting enormous interest due to their unique properties, including a great variety of sizes and shapes, and their potential applications in pigments, catalytic processes, (gas) sensors, electrochemical devices, magnetic materials, and energy-storage systems [5,6]. Therefore, the potential negative health and eco-toxic effects of cobalt oxide NPs (Co-NPs) have attracted considerable attention because of their accidental or intentional release into the environment (e.g., accidental occupational inhalation of Co-NPs by workers in a nuclear power plant) [7]. Human exposure to cobalt can also occur in the works dealing with cobalt-bearing materials or in dental clinics during after joint replacement of the cobalt–chrome alloy in implants [8]. The exposure to cobalt and its compounds in these conditions can cause various lung diseases, including interstitial pneumonitis, fibrosis, and asthma [9].

Although the precise toxicity mechanisms of NPs remain largely unknown, studies have shown that the toxicity of NPs, when introduced to ecosystems, is often caused by their unique physical and chemical characteristics (e.g., particle size, surface characteristics, shape, and reactivity) [10,11,12]. For instance, the toxicity of Co-NPs may be characterized either by their direct uptake by cells or by their solubilized metal ions in the media [13]. Monikh et al. [14] observed the combined effect of shape, size, and ecocorona controls the Au-NP’s attachment and physical toxicity to cells. The possible toxic effects caused by NPs include synthesis of inflammatory mediators, damage to cell membranes, damage to DNA, and altered cellular redox balance toward oxidative stress, which causes abnormal cell functioning or cell death [15]. Oxidative stress is a normal cellular process that occurs due to a lack of the ability of biological systems to detoxify reactive oxygen species (ROS), which are reactive intermediates, or repair the resulting damage [16]. However, there is a paucity of previous works on the effects of cobalt on organisms, particularly, considering nano-sized fine particles containing cobalt.

Various whole organisms (e.g., bacteria, algae, protozoa, plants, and fish), as well as their specific metabolic processes (e.g., enzyme activity, enzyme biosynthesis, and bioluminescence activity), have been adopted to evaluate the toxicity of chemicals in environmental systems [2,4]. Research into the toxicity of NPs has mostly demonstrated different levels of effects depending on the organisms considered. For example, TiO_2_ and ZnO NPs, which are used widely in sun-care products and self-cleaning coatings, have been shown to have toxic effects and can inhibit the growth of microalgae, crustaceans, and bacteria. However, other studies have shown opposite results [17,18]. Among the various conditions of bioassays, an understanding of each test organism’s sensitivity is important in evaluating the toxicity of contaminants. The effects of plant processes on seed germination are considered rapid responses to acute toxicity and are particularly relevant when phytotoxic contaminants are present in soil. On the other hand, evaluation of plant biomass and root/shoot elongation in environmental biomonitoring has found both acute and chronic toxicity [19]. Algal growth is also commonly used to examine the toxicity of NPs in water systems. A short-term bacterial reverse mutation assay, the Ames test, has also been used to detect a wide range of chemical substances that can inflict genetic damage, leading to gene mutations [20].

The single-cell gel electrophoresis assay (comet test) is also one of the most frequently used genotoxicity tests for NPs [21]. Because NPs are important global environmental contaminants, it is important to select suitable biological activities for their risk assessment. In addition, a single bioassay (organism or endpoint) never provides an appropriate estimate of toxicity; therefore, a combination of several bioassays is necessary to provide an accurate bioassessment of contaminants in environments. Additionally, as the effects of NPs may vary depending on the testing conditions, these also need to be given attention for proper assessment.

The main purpose of this study was to assess and compare the potential toxic effects of two groups of commercially available Co-NPs of different size (type A and type B). To achieve this goal, nine biological activities were adopted. These included the bioluminescence activity of *E. coli* recombinant strain (RB1436), the enzymatic and biosynthesis activity of β–galactosidase produced by *E. coli*, bioluminescence-producing bioreporter strain (*xyl-lux* gene; *Pseudomonas putida mt-2* KG1206) activity, bacterial gene mutation of *Salmonella typhimurium*, algal (*Chlorella vulgaris*) growth, and germination and root/shoot growth of genus *Lactuca*. The characteristics of the tested Co-NPs were also determined using various instruments and compared with those indicated by the manufacturer.

## 2. Results

### 2.1. Characterization of the Tested Co-NPs

The morphological characteristics of the two types of Co-NPs tested were investigated by SEM and TEM and are shown in Figure 1 and Figure 2, respectively. As shown in the SEM photographs of Figure 1, the type A Co-NPs were mostly roughly spherical aggregates with a size of 10–30 nm, with some aggregates larger than 50 nm, and of relatively homogeneous shape and size. Similar features are also evident in the TEM photographs of Figure 2, showing particles roughly the same size as the Co-NPs presented by the manufacturer. On the other hand, the type B Co-NPs were clustered into somewhat angular shapes that were distorted. The type B Co-NPs were mostly 80–150 nm, but many over 200 nm or between 500 and 1000 nm were also observed, as shown in Figure 1 and Figure 2. Figure 3 shows the XRD patterns of the two different sizes of Co-NPs (type A and type B). The diffraction peaks of the two types of cobalt oxide (Co_3_O_4_) for (111), (220), (311), (222), (400), (422), (511), and (440) planes were observed easily at 2θ values of 19.01°, 31.31°, 36.85°, 44.81°, 55.85°, and 65.39°. As can be seen in Figure 3, the intensity of the peaks of type A was higher than that of type B, indicating increased crystallinity with a larger crystallite size. The sizes of cobalt oxide crystallites were calculated from the most intense peak (311) at 2θ = 36.26° using the Debye–Scherer equation: $D_{XRD}=\frac{0.9\lambda }{\beta \cos\theta }$, where *D_XRD_* is the crystallite size (nm), λ is the wavelength of Cu Kα radiation, β is the full width at half maximum of the diffraction peak, and θ is the Bragg’s angle. The average sizes of type A and type B crystallites, calculated from the equation, were approximately 26 nm and 69 nm, respectively. These results are considered to be consistent with the Co-NP sizes provided by the manufacturer (Table 1). The Fourier-transform infrared spectroscopy (FT-IR) spectra of the Co-NPs of the two different sizes (types A and B) were used in the study (Figure 4). The two representative characteristic absorption bands of Co-NPs in the FT-IR spectra were at about 663.4–667.3 (ν_1_) and 568.9–578.6 (ν_2_) cm^−1^ indicating the stretching vibrations of the Co-O bands [22]. The intense broad band of 3440 cm^−1^ and the weak band of 1635 cm^−1^ were attributed to O-H stretching vibration and the bending modes of the adsorbed water molecules in the FT-IR spectra of the type A Co-NPs, respectively. The absence of two peaks in the FT-IR spectra of type B was thought to be due to a decrease in water content due to increased calcination temperature during the Co-NP manufacturing process [22,23,24]. The broad absorption band at 3440 cm^−1^ and weak bands were observed only in the FT-IR spectra of the type A Co-NPs. The specific surface areas were determined by nitrogen gas (N_2_) adsorption–desorption measurements. The specific surface areas of the Co-NPs calculated by the BET method for types A and B were 38.94 ± 0.18 m^2^/g and 0.929 ± 0.006 m^2^/g, respectively. Assuming that Co-NPs are spherical and have similar sizes, their average particle size can be obtained from the equation $D_{BET}=\frac{6000}{\rho \cdot S_{BET}}\;$, where *D_BET_* is the average diameter of the nanoparticle, *S_BET_* indicates the measured specific surface area of the Co-NPs in m^2^/g, and ρ is the theoretical density of cobalt oxide of 6.11 g/m^3^. The calculated average sizes of type A and type B Co-NPs were 25.3 nm and 1062 nm, respectively.

### 2.2. Effects of Co-NPs on the Activities of Various Bacterial Systems

The effects of Co-NP size on the bioluminescence activity of strain RB1436 was determined at 100 and 200 mg/L for both sizes of particle. During growth, the control produced a mean of approximately 300 RLU of bioluminescence after 1 and 1.5 h of incubation, depending on the experimental conditions. The relative effects of the two Co-NP sizes on bioluminescence activity during incubation are shown in Figure 5. The relative activities after 30 min of exposure to the type A Co-NPs were 45% and 2.1% of the control levels at 100 and 200 mg/L, whereas exposure to type B Co-NPs resulted in relative activities of 40% and 66% of the control levels at 100 and 200 mg/L, respectively (Figure 5A). No considerable reductions in bioluminescence activity were observed following these exposure periods. Toxicity was calculated based on the mean bioluminescence activity after 1 h and 1.5 h of exposure. At 100 and 200 mg/L of NPs, the observed bioluminescence inhibition was 51 ± 1.3% and 98 ± 0.2% in the presence of 10–30 nm NPs, whereas it was 40 ± 2.5% and 66 ± 2.2% in the presence of 50–80 nm NPs, respectively (Figure 5B). Small particles (type A) showed approximately 1.2- to 1.5-fold greater toxicity to bioluminescence activity compared to large particles (type B) (*p* = 0.0001–0.0070). The effects of the Co-NPs on another bioluminescence process (bioreporter strain KG1206) were also investigated at 5 mg/L and 10 mg/L for both sizes of Co-NPs (Figure 6). During the 3 h incubation period, the maximum activity was at 1.5 h of incubation, followed by rapid decreases in bioluminescence activity (Figure 6A). The maximum bioluminescence activities at 1.5 h of incubation at 5 mg/L and 10 mg/L of 10–30 nm Co-NPs were 793 ± 99.2 RLU and 185 ± 43.5 RLU, which correspond to 49% and 11% of the control activity, respectively. However, at 5 mg/L and 10 mg/L of type A Co-NPs, the maximum bioluminescence activities were 637 ± 100.6 RLU and 404 ± 129.6 RLU, equivalent to 56% and 35% of the control activity, respectively (Figure 6A). Comparisons under each condition were made based on the sum of the bioluminescence, which was measured at 30-min intervals during the 3-h incubation. The control produced approximately 2000 RLU of total bioluminescence. As the result of a comparison between the relative activities based on the total bioluminescence, 52 ± 2.0% and 18 ± 2.4% activities were observed at 5 and 10 mg/L of the type A Co-NPs, respectively. However, the relative activities at the same concentrations for the type B Co-NPs were 59 ± 10.9% and 53 ± 7.7% (Figure 6B). No significant differences were observed for the exposure of 5 mg/L (*p* = 0.4874). However, notably, exposure to 10 mg/L of small Co-NPs (type A) produced a 1.8-fold increase in toxicity on bioluminescence activity compared to exposure to large Co-NPs (*p* = 0.0022). Under the tested conditions, the toxic effects of the Co-NPs on the bioluminescence activity of KG 1206 were ranked as follows: type A (10 mg/L) > type A (5 mg/L) > type B (10 mg/L), type B (5 mg/L).

The effects on the enzymatic activity and biosynthesis of β-galactosidase were examined by measuring the color changes of the mixture (chromogen and enzyme) solutions at two concentrations (200 and 600 mg/L) of Co-NPs (Figure 7). The average absorbance of the control (no Co-NP exposure) after 40 min of incubation was 1.77 ± 0.044, depending on the batch set of enzymatic activity. No stimulation of enzymatic activity was observed under the conditions tested. At 200 mg/L and 600 mg/L, exposure to type A Co-NPs resulted in enzyme activities of 76 ± 14.2% (corresponding to a toxicity of 24%) and 15 ± 1.7% (corresponding to a toxicity of 85%) compared to the control levels, respectively (Figure 7A). In contrast, relatively low inhibition of enzymatic activity was observed with type B NPs, with relative enzymatic activities of 87 ± 2.6% and 64 ± 7.5% compared to the control at 200 mg/L and 600 mg/L of Co-NP, respectively. The differences in the effects on the biosynthesis of β-galactosidase were also determined after 1 h of exposure of the Co-NPs to the biosynthesis process. The absorbance without Co-NP exposure ranged from 1.661–1.740, whereas it ranged from 0.896–1.366 with Co-NP exposure, depending on the conditions tested. Figure 7B shows the changes in relative biosynthetic activity (%) at the tested concentration doses of the two sizes of Co-NPs. No stimulation of biosynthetic activity was observed under the conditions examined. At 200-mg/L and 600-mg/L exposures of type A Co-NPs, the biosynthetic activities were 65 ± 1.7% (corresponding to a toxicity of 35%) and 58 ± 5.1% (corresponding to a toxicity of 45%) of the control levels, whereas activities of 74 ± 5.2% (corresponding to a toxicity of 26%) and 70 ± 1.1% (corresponding to a toxicity of 30%) occurred upon exposure to type B Co-NPs, respectively. Under the tested conditions, the toxic effects of Co-NPs on both enzymatic activity and biosynthetic process were ranked as follows: type A (200 mg/L) > type A (100 mg/L) > type B (200 mg/L) > type B (100 mg/L). In the case of the enzymatic activity and biosynthesis of β-galactosidase, small Co-NPs showed a 1.9–2.4-times and 1.4-times higher inhibitory effect compared to large Co-NPs under the tested conditions, respectively (*p* < 0.0191).

### 2.3. Effects of Co-NPs on Gene Mutation

The potential mutagenic differences were evaluated by employing the mutant bacterial strain *S. typhimurium* TA 98 at two concentrations (100 and 500 mg/L). The spontaneous reversion of TA 98 ranged from 19 to 23 (20 ± 2.3) colonies. For the positive control tests, 2-nitrofluorene (2.5 μg/L) was chosen based on the preliminary results. At 100 mg/L and 500 mg/L, colonies formed by reverse mutation showed 27 ± 4.9 CFU and 25 ± 1.2 CFU (corresponding to a toxicity 35% and 25%) for the small type A Co-NPs and 16 ± 5.1 CFU and 15 ± 6.4 CFU (corresponding to a toxicity −20% and −29%) for the large type B Co-NPs, respectively (Figure 8A and Appendix AA). The mutagenic ratios (MRs) of the tested NPs ranged from 0.7 to 1.3. The MRs of the type A Co-NPs were 1.3 ± 0.24 and 1.2 ± 0.06 at 100 mg/L and 500 mg/L, respectively. In the presence of type B Co-NPs, the MR values were 0.8 ± 0.25 and 0.7 ± 0.32 at 100 mg/L and 500 mg/L, respectively. The statistical differences in MRs or colonies formed by reverse mutation observed between the two sizes of Co-NPs tested (*p* = 0.0194) indicated a higher mutation rate for the small Co-NPs than for the large ones. The revertant gene mutation of small Co-NPs showed a 1.7-fold increase in colonies compared to that of the large Co-NPs in the test.

### 2.4. Effects of Co-NPs on Algal Growth

The effects of the two different sizes of Co-NPs on the activity of algal growth were evaluated at two concentrations (1000 and 2000 mg/L) by measuring cell count and chlorophyll content. The cell count and chlorophyll content of algal cultures without Co-NP exposure were 8.2 × 10^6^/mL and 9.73 g/m^3^, respectively. Depending on the particle size and exposure concentration, relative cell counts and chlorophyll contents ranged from 1.5–51.2% and 20.3–67.9% compared to the control, respectively (Figure 8B). The effect of chlorophyll content showed 2.76-times higher activity (corresponds to lower sensitivity and lower toxicity) compared to that of cell count, with average activities of chlorophyll content and cell count of 39.6% and 14.9% under the tested conditions, respectively. Based on the mean toxicity value of all sets, the toxicity to cell count (total average 85.0%) was observed to be about 1.4-fold higher than the chlorophyll content toxicity (total average 60.4%) (Appendix AB). Among the tested conditions, a great toxic effect on the activity of algal growth of 98.5–97% toxicity for cell count compared to the control levels was observed at 2000 mg/L for both of the tested sizes. At the 1000 mg/L and 2000 mg/L exposure conditions, the toxicity caused by small particles was similar, with 96% and 98.5% for cell count and 74.8% and 79.7% for chlorophyll content, whereas considerable differences were observed with large particles, which produced 48.8% and 97% toxicity for cell count and 32.1% and 56.6% toxicity for chlorophyll content, respectively (Appendix A). Comparing the mean effects of the two different particle sizes, small Co-NPs showed an approximately 1.5-fold increase in toxicity compared to large Co-NPs. In particular, significant statistical differences in toxicity were observed between the different sizes for both observed endpoints at the low concentration (1000 mg/L) tested, showing nearly 2.1-times higher algal growth toxicity of small NPs compared to that of large NPs (*p* = 0.0117). With respect to both endpoints, the average toxicity ranking was as follows: type A (2000 mg/L) > type A (1000 mg/L) > type B (2000 mg/L) > type B (1000 mg/L).

### 2.5. Effects of Co-NPs on Seed Germination and Root/Shoot Growth

The effects of the two different particle sizes of Co-NPs were evaluated on the basis of changes in germination and root/shoot growth of the genus *Lactuca* at 1000 and 2000 mg/L. For the control, an average of 13–17 seeds per batch of 20 seeds germinated (longer than 2 cm growth) after a 3-day incubation period. Germination activity was inhibited slightly by small Co-NPs, which showed 84 ± 31.9% and 79 ± 23.7% of the control levels at 1000 mg/L and 2000 mg/L, respectively (Figure 9A). However, slight stimulation of germination activity, 105 ± 14.9% and 103 ± 7.2% of the control germination levels, was observed with the large Co-NPs (type B) at 1000 mg/L and 2000 mg/L, respectively (Figure 9A). Statistically considerable differences were also observed at the exposure levels of 1000 mg/L (*p* = 0.0409) and 2000 mg/L (*p* = 0.0045) between the two sizes of particles. Other vegetative-response endpoints (root/shoot growth) were also determined to study the effects of Co-NP particle size. After 4 days of incubation, the root and shoot lengths of the control group were 70 ± 25.3 mm and 15 ± 2.2 mm, respectively. In the experimental groups, the RRLs of the Co-NPs at 1000 and 2000 mg/L were 51 ± 5.4% and 37 ± 3.4% for the type A Co-NPs and 85 ± 1.0% and 67 ± 6.6% for the type B Co-NPs compared to the control, respectively (Figure 9B). Significant inhibition of root growth, approximately 0.37 to 0.85 times the control root length, was observed compared to the other endpoints. In contrast to the root growth (RRL of 37–85% of the control activity), stimulation of shoot growth, showing RSL from 146–196% of the control levels, occurred under all tested conditions. There was about 3.3- and 2.0-times higher root growth toxicity for the small particles compared to that of the large particles at 1000 mg/L (*p* = 0.0017) and 2000 mg/L (*p* = 0.0012), respectively. The average toxicity of the root growth observations of small Co-NPs (57% toxicity) was approximately 2.4 times greater than that of large Co-NPs (24% toxicity). The toxicity rankings on germination and root growth were as follows: type A (2000 mg/L) > type A (1000 mg/L) > type B (2000 mg/L) > type B (1000 mg/L).

## 3. Discussion

### 3.1. Characterization of Tested Co-NPs

Various physicochemical characteristics of NPs and environmental factors have been reported to affect their toxicity, fate, and bioavailability in the environment [3,15,25,26]. Among these factors, the size of an NP is an important factor in determining how it interacts, distributes, and accumulates in organisms [15]. Prior to evaluating the effects of different sizes of Co-NPs on biological activities, the structural and morphological characteristics of the purchased ones were determined using SEM, TEM, XRD, FT-IR, and a surface-area analyzer. One distinctive feature was that distorted hexagonal NPs, which were related to the spinel structure of the Co-NP, were observed in TEM images of the type B Co-NPs. From the SEM and TEM observations, the images show the presence of some large Co-NPs, which could be the result of the aggregation or overlapping of small NPs. Especially, the type B Co-NPs were substantially larger than the manufacturer’s proposed size of 50–80 nm, unlike the case for type A. The large difference in size between the SEM and TEM observations and the manufacturer’s value is thought to be due to the difference between the particles and the crystallite. A nanoparticle by SEM observation is thought to be an agglomerate composed of two or more individual crystallites, such that the size of the nanoparticle has a value greater than the size of an individual crystallite determined by the Debye–Scherer equation from the XRD pattern. All of the diffraction peaks appearing in the XRD patterns match very well with those reported in the literature [23,24]. No other peaks were detected in the XRD patterns, indicating that the tested Co-NPs had high purity. In the FT-IR spectra, the former absorption band (663.4–667.3 cm^−1^) is attributed to the stretching vibrations of the Co–O bands in the octahedral site and the latter one (568.9–578.6 cm^−1^) is characteristic of the Co–O vibration band in the tetrahedral site of the Co_3_O_4_ [24]. The characteristic absorption bands in the FT-IR spectra indicated the pure Co_3_O_4_ spinel phase. The broad absorption and weak bands only in the FT-IR spectra of the type A Co-NPs indicated the effects of O–H stretching and the bending mode of water molecules [22]. Comparing the measured values of specific surface area to those provided by the manufacturer, the type A Co-NPs showed relatively similar values, but the type B Co-NPs had much smaller values of about 1/10. This is thought to be related to the increase in particle size of the type B Co-NPs that is evident in the SEM and TEM images. The size of the type A Co-NPs calculated from the specific surface area measurement was relatively consistent with the manufacturer’s value, XRD result, and SEM/TEM observations. However, the size results for type B were different from the other experimental results, which is thought to be closely correlated to the formation of nanoparticle aggregates, as observed in the SEM and TEM images.

### 3.2. Effects on Five Bacterial Activities

The study of bacterial activity is widely used to determine the acute effects of various environmental contaminants. In this study, the effects of the particle size of the Co-NPs was evaluated based on five bacterial activities (two bioluminescence activities, an enzymatic activity, a biosynthesis of enzyme activity, and a gene mutation activity) at two concentrations for each activity, which were determined based on the preliminary tests. The effects of the Co-NPs varied depending on the bacterial strain (or endpoint), concentration, and particle size. Although the effects of Co-NPs may differ depending on the tested conditions, small Co-NPs showed statistically higher toxicity compared to large Co-NPs under experimental conditions for all tested bacterial systems. Different effects were also observed on the systems of two bacterial bioluminescence processes tested (strains RB1436 and KG1206), with greater inhibitory activity against KG1206 than against RB1436. However, for both bioluminescence activities, small particles showed a total average of 1.44 times higher inhibitory effect compared to large particles under the tested conditions. Of the various properties of NPs that affect bacterial activity, solubilized metal ions from MNPs may be responsible for the observed inhibitory effects [27]. Positively charged metal ions from NPs would likely interact on the cell surface of Gram-negative bacteria, which is negatively charged at near-neutral pH due to the presence of lipopolysaccharides, potentially leading to inhibition of enzyme biosynthesis [28]. In this study, soluble cobalt concentrations were measured for bacterial bioluminescence strain experiments to determine the contribution of soluble metal concentrations on bacterial activity. A very low concentration of dissolved cobalt (4–5 μg/L) was detected at 5 and 10 mg/L Co-NPs. Therefore, the contributions of soluble cobalt on bacterial activity were thought to be minimal in this study. Other studies have also reported that the effects of dissolved metal ions do not account for the total observed toxicity [29]. Similar to the results of this study, Papis et al. [30] found that the number of cobalt ions released by Co-NPs in the culture medium was very low and did not reach any significant concentrations. Therefore, the toxicity of NPs might be mostly influenced by the particle characteristics, rather than by released metal ions; however, this would depend on the test conditions. Heinlaan et al. [17] also suggested that the toxic effects are caused mainly by the intimate contact between cell and particle, because bacteria are largely protected against NP entry (no transport mechanisms for supramolecular and colloidal particles). However, the relative contributions of the particles and solubilized ions of metal-based NPs are not well described at present and these may behave differently [31,32]. In addition, the toxicity behavior of NPs on the single bacterial strain is far from the practical conditions of polymicrobial communities, which may weaken the toxic effect of NPs [33].

The potential mutagenicity and mutagenic differences between the two sizes of Co-NPs were evaluated by employing the mutant bacterial strain *S. typhimurium* TA 98 (Ames test). The spontaneous reversion of TA 98 in this study was slightly lower than previously reported results [29]. As in other bacterial activities, the small Co-NPs induced higher toxicity (higher mutation rates) than the large Co-NPs (Appendix AA). However, the MRs were all lower than 2, suggesting the possibility of no mutagenic effects under the conditions examined in this study (ratio higher than 2.0 indicates positive mutagenicity). Magaye et al. [8] reported that the exact mechanisms of Co-NP-induced carcinogenesis in experimental animals are not clear, but they did confirm that enhanced oxidative stress, inflammatory response, and abnormal apoptosis may play major roles in the carcinogenicity. Guan et al. [34] reported that the changes in bacterial transcriptional regulation in the presence of NPs reflected the disturbance in the physiological activities and loss of cell integrity, leading to damage of bacterial cells or death. According to another study, bacterial mutagenicity assays (Ames test) may not be suitable for measuring mutations induced by NPs because NPs may not be able to diffuse across bacterial cell walls [35]. This lack of uptake ability of bacteria could potentially give false results, so the actual uptake of NPs into bacterial cells needs to be determined. This information indicates that one genotoxicity assay is insufficient to cover all potential forms of genetic mutation potentially caused by NPs and implies that the results of various methods need to be integrated for the proper mutagenicity.

### 3.3. Effects on Algal Growth

The effects of Co-NPs on the activity of algal growth were evaluated by measuring two endpoints (cell count and chlorophyll content). The absorbance-based effect was not considered because of the color interference of Co-NPs in solution. No growth stimulation was observed under tested conditions. Similar to the effects on all bacterial activities, the effect of small particles on algal growth was considerably greater than that of large particles, especially at the low concentration tested (1000 mg/L) (Appendix AB). Researchers have previously reported that NPs can have direct or indirect effects on algal growth [36,37]. For example, direct toxic effects are mainly determined by chemical composition and surface reactivity due to the great surface area per mass, whereas indirect effects are caused by the release of toxic ions or by physical restraints. The cell wall, composed of carbohydrates (cellulose) linked to form a multi-sheath rigid complex and proteins, may be an initial site for interactions (direct effect) and a barrier to the intake of NPs into algal cells, which may vary depending on the structure [38]. Algae cell walls are capable of sieving out NPs in the environment, and there is a possibility that only particles smaller than 20 nm are likely to enter the cytoplasm after passing through the cell membrane, and continuously interact with the phospholipid bilayer, damaging the membrane structure [14,37]. The higher inhibition of the smaller Co-NPs (type A) compared to the larger Co-NPs (type B) on algal growth in this study may have been due to differences in the passage through cell walls. Hund-Rinke and Simon [39] also reported that the toxicity to the green algae was clearly dependent on the specific surface area (particle size) of TiO_2_ NPs exposed. However, many researchers have suggested that various factors affect the growth of algae as a result of environmental exposure to MNPs: the release of soluble metal ions, reactivity of MNPs to photosynthetic enzymes, adhesion of algal cells leading to disruption of the cell membrane or reduction in cellular nutrient uptake, and generation of photocatalytic ROS NPs [40]. Pikula et al. [41] also suggested that the toxicity of different algal species to NPs depends on the ability to interact with the components of the microalgal cell wall.

### 3.4. Effects on Germination and Root/Shoot Growth

Assessments of the germination and root/shoot growth of seeds are also used widely for acute toxicity measurements because of their sensitivity, simplicity, low cost, and ease of handling [42]. Slight inhibition of germination activity was observed with small Co-NPs, while slight stimulation with large Co-NPs. Statistically significant differences of toxic effects were also observed between the two particle sizes tested. An important influencing factor for seed germination is the extent to which particles penetrate the embryonic tissue via the seed coat, which varies depending on the seed species and physicochemical properties [43]. NPs able to reach the embryonic tissue may inhibit germination by affecting the starch-degrading enzymes as well as other activities required for seed germination. Comparing the effects on the root growth, statistically significant differences in toxicity were also observed between the different sizes for both concentrations tested. Very low concentrations of soluble cobalt metal were also observed in this experimental solution for root growth effects, i.e., 64 ± 14 μg/L and 439 ± 135 μg/L for the small particles (type A) and 5 ± 0.6 μg/L and 8 ± 1 μg/L for the large particles (type B) at 1000 mg/L and 2000 mg/L Co-NP, respectively. As such, the effect by soluble metal on root growth would be insignificant, and the main influence would be by the physical effects of contact, adsorption, and uptake of particles to the root tissues and the influence on metabolic processes in the cytoplasm following uptake. In this study, root growth was most sensitive to Co-NPs, followed by seed germination and shoot growth. The lack of inhibitory effects on shoot growth compared to the other endpoints may be due to the absence of contact with particles. As for the other biological activities investigated in this study, the small Co-NPs showed a higher inhibitory effect on plant activity than did the large ones, possibly due to their high specific surface area. Other researchers have also reported that small NPs exhibit greater inhibitory effects on root growth than larger ones [44]. Lee et al. [45] reported that the toxicities of mung bean and wheat were attributable to the Cu-NPs rather than the dissolved Cu ions (0.3 mg Cu ions/L in 1000 mg/L Cu-NPs). A similar discussion was also provided regarding the phytotoxicity of ZnO NPs [27]. Srivastave [46] reported that the treatment of Co-NPs on root meristems showed mitotoxic and genotoxic effects and it causes a reduction in the active mitotic index and also disturbed normal mitotic behavior of cells by inducing various types of chromosomal aberrations. Although the phytotoxicity of NPs to plants remains largely unknown, several studies have reported that toxicity can be affected by several factors, including solubilized metal ions, partial solubility, intimate contact ability, NP uptake and accumulation, production of ROS upon contact, NP type, morphology (shape), particle size, crystallinity, surface chemistry, residual chemical impurities, and environmental factors such as pH and ionic strength [47,48,49,50].

### 3.5. Overall Effects on the Biological Activities

Considering the overall effects on the biological activities investigated in this study, the toxic effect of the small Co-NPs (53 ± 5.0%) showed approximately 2.1-times higher than that of the large Co-NPs (25 ± 5.6.%) (Table 2). In general, the sensitivities and toxic mechanisms vary depending on the organism tested [48]. Therefore, combinations of results of various biological activities, rather than only a single activity, would better facilitate accurate assessment of NPs’ toxicity in ecosystems. Although the precise contribution of the effects of different sizes of NPs on many biological activities is unknown, this study indicated that the effect of NPs on biological activity is thought to be mediated generally by the physical characteristics rather than the soluble metals. However, the implication of multiple applications of NPs is needed to provide more realistic approaches [51]. In addition, NPs are likely to react with the constituents of environmental matrices, such as soil, and the resulting adsorption and aggregation reactions will modify their properties with respect to mobility, bioavailability, exposure time, and ecotoxicity. 

## 4. Materials and Methods

### 4.1. Characterization of NPs, Chemicals, and Determination of Cobalt in Solution

Two different sizes of 99% Co_3_O_4_ NPs (Co-NPs: types A and B) were obtained from NanoAmor (Nanostructured & Amorphous Materials Inc., Houston, TX, USA) and used for this research (Table 1). The Co-NPs were suspended directly in deionized water and dispersed by ultrasonic vibration for 10 min prior to use. All other chemicals were reagent grade chemicals purchased from Sigma Chemical Co. or Aldrich Chemical Co. (St. Louis, MO, USA). The solution samples at the end of the incubation period of the tests were filtered (0.45 μm) to determine the concentration of dissolved cobalt ions using an inductively coupled plasma optical emission spectrometer (Optima 7300DV ICP-OES, Perkin-Elmer Inc., USA).

The Co-NPs tested were characterized using a high-resolution field emission scanning electron microscope (FE-SEM, Hitachi S-4800, Japan) and an ultra-corrected energy-filtered transmission electron microscope (UC-EF-TEM, LIBRA MC, Carl Zeiss, Germany). X-ray diffraction (XRD) and Fourier-transform infrared spectroscopy (FT-IR) analyses were performed using the two different size groups of Co-NPs. The average particle size of the Co-NP samples was calculated from the XRD patterns by applying the full-width at half-maximum (FWHM) of the characteristic peak to the Scherer equation. FT-IR spectra for the Co-NPs were measured in the wavenumber range of 4000–400 cm^−1^ at a spectral resolution of 4 cm^−1^. The correction spectra of each Co-NP were obtained by subtracting the background interference spectra, and then the average chart was taken as a last Co-NP sample spectrum. The specific surface area (BET surface area) and average pore diameter of the Co-NPs were determined by the nitrogen (N_2_) adsorption method at a bath temperature of 77.3 K using a Micromeritics ASAP 2010 Chemisorption Surface Area Analyzer (Micromeritics, Norcross, GA, USA).

### 4.2. Toxicity Bioassay Assessment

Toxicity was evaluated based on nine biological activities: five bacterial activities, algal growth activity, and three activities of a plant (*Lactuca*). Two concentrations tested for each bioassay are provided in Appendix A. The statistical analysis for the experimental groups was performed using Student’s *t*-test (http://www.graphpad.com). The experimental values were compared to their corresponding controls to evaluate the differences between the pairs, where they are considered statistically significant if the p-value is less than 0.05.

### 4.3. Effects of Co-NPs on Bacterial Bioluminescence Activity

Two different bacterial bioluminescence processes were adopted for this investigation. For the first, the *E. coli* DH5α RB1436 (called RB1436 hereafter) mutant strain containing a plasmid subjected to a spontaneous deletion event upstream of the *lux*CDABE genes was used. During growth, a constitutive promoter to express the *lux* genes allows the production of bioluminescence [52]. For the bioassay, 1 mL of the diluted bacterial suspension solution mixed with 9 mL of Co-NP solution and then bioluminescence activity measured at 1 and 1.5 h during the incubation period was used for the toxicity evaluation. For the second, the bioreporter strain *Pseudomonas putida* mt-2 KG1206 (called KG1206 hereafter), containing the intact TOL plasmid and a *Pm-lux* fusion plasmid, was used. This strain produces bioluminescence in the presence of toluene analogs and intermediates [53]. To determine the effects of the Co-NPs, an inducer chemical solution (final 1 mM *o*-chlorotoluene) was added to the mixture solution of culture and sample. The bioluminescence was measured in triplicate during the incubation period with a Turner 20/20 luminometer (Turner Design Inc., San Jose, CA, USA) with a maximum detection limit of 9,999 relative light units (RLU).

### 4.4. Effects of Co-NPs on Enzymatic Activity and Enzyme Biosynthesis

The *E. coli* EW1b mutant strain (provided by the University of Florida) was used for the production of β-galactosidase enzyme. The strain in lactose broth (LB) medium (35 °C, 100 rpm) was allowed to grow until the optical density (OD_550_) reached approximately 0.7. Isopropyl-β-D-thiogalacto-pyranoside (IPTG, Sigma Chemical Co., USA; final concentration, 100 mg/L) was added and left for 4 h for enzyme induction. The culture solution was centrifuged and washed twice. The washed pellet was re-suspended in 1% sterilized trehalose and distributed into vials (3–5 mL per vial). This suspension solution was deep-frozen and used to examine the effects of Co-NPs on enzyme activity. In this test protocol, a small volume (0.1 mL) of the enzyme solution was directly exposed to the Co-NP solution (0.9 mL) for 60 min at 35 °C. The mixture solution (0.2 mL) was transferred to a 96-well plate with chromogen solution added (0.1 mL) with chlorophenolred-β-D-galactopyranoside (CPRG, Boehringer Mannheim GmbH Germany; 125 mg/L in 0.15 M p-buffer, pH 7.4) and incubated until the color changed from yellow to red (purple).

Based on the results of several preliminary tests, the following procedure was adopted for the evaluation of the effects of the Co-NPs on enzyme biosynthesis. A mixture of the culture (0.1 mL) and IPTG (0.1 mL) (final concentration 100 mg/L) was exposed to 0.9 mL of the sample (Co-NP solution) for 60 min at 35 °C at 100 rpm. To stop the biosynthesis reaction, 0.1 mL of 0.5 M Na_2_CO_3_ solution was added to the mixture solution. Mixture volumes of 0.2 mL and 0.1 mL of CPRG were pipetted into all wells of a 96-well plate and incubated at 35 °C until the color changed. Both activities were determined by measuring the absorbance (OD_577_) using a 96-well microplate reader (MB-MAC-29, Maxim Bio Co., Italy).

### 4.5. Effects of Co-NPs on Bacterial Gene Mutation

The plate incorporation procedure with *Salmonella typhimurium* TA 98 was used for a mutagenicity test and the strains were maintained and stored using standard methods [20]. A control of 0.05 mL DMSO or 0.05 mL sample in 2 mL top agar (6 g NaCl, 6 g agar, 100 mL 0.5 mM histidine/biotin solution, and 900 mL water) was placed onto a GM agar plate with a mixture of 0.1 mL culture and 0.5 mL buffer. Each sample was plated in triplicate, and his^+^ revertants were counted after 48 h of incubation at 37 °C. 2-nitrofluorene of 2.5 µg per plate was used as a positive control to evaluate the sensitivity of the bacterial strains to Co-NPs.

### 4.6. Effects of Co-NPs on Algal Growth Activity

The effects of the Co-NPs on algal growth were evaluated using the green algae species *Chlorella vulgaris* (KCTC AG10002) obtained from the Korean Culture and Tissue Collection (KCTC). The algae were cultured for 3 days in BG-11 medium at 30 °C and 150 rpm under 5000 lux, and diluted to a final OD_730_ of 0.3 for testing. An amount of 1 mL of the NP solution was exposed to 19 mL of the algal culture (3 days at 30 °C and 150 rpm) for the NP inhibition test. Growth inhibition was determined after 3 days of incubation by assessing by measuring the cell count and chlorophyll content. Chlorophyll extracted with 90% (*v/v*) acetone was measured using a UV/VIS spectrophotometer (Shimadzu 1240 UV mini, Seoul, Korea), and the chlorophyll content was calculated using previously reported equations [54,55]. The algal cells were counted using a counting chamber (Marienfeld, Lauda-Königshofen, Germany) under an optical microscope.

### 4.7. Effects of Co-NPs on Seed Germination and Root/Shoot Growth

Seeds (*Lactuca sativa L.*) were obtained from a local seed store, and were distributed by a commercial seed company (Nongwoo Bio., Korea). For the germination test, all seeds were soaked in 3% H_2_O_2_ for 10 min to sterilize their surfaces and rinsed with distilled water. Filter papers were then placed in a sterilized Petri dish and moistened with 5 mL of aqueous solution containing Co-NPs and distilled water for test samples and controls, respectively. Twenty seeds of each species were incubated in the dark at 23 ± 2 °C in a plate covered with a lid. Germinated seeds were counted after 3 days of incubation. To evaluate the effects of Co-NPs on root and shoot growth, germinated seeds were transferred to serum vials containing 30 mL of test solution. After incubation of the vials for 4 days at 25 °C and 70 rpm, the root length and shoot height of seedlings were measured from their junctions to the longest tip. The measured growths of root and shoot in test solutions were expressed as the percentage inhibition (%) of the relative root length (RRL) and relative shoot length (RSL) compared with those of the controls.

## 5. Conclusions

In summary, this study demonstrated that each of the biological activities considered has a different level of sensitivity to different particle sizes of Co-NPs. This result implies that the adoption of a wide range of biological activities, as well as information for particle sizes, may constitute a better tool for accurately assessing the toxicity of NPs in the environment. Overall, the results observed under the tested conditions indicated a significant inhibitory effect on all bacterial activities followed by algal growth and plant activities (except for shoot growth). In particular, the experimental results clearly confirm that the small Co-NPs had a consistently higher inhibitory effect compared to the large Co-NPs under all conditions tested. The toxicity of Co-NPs induced by released ions might be insignificant and mainly affected by the particle physical characteristics. However, the (eco) toxicity of NPs could be quite variable depending on laboratory and natural environmental conditions, as well as on the combined effect of physicochemical properties of NPs. Therefore, there is a need for long-term and real-time assessments of different physicochemical properties of NPs for the safe and responsible use of NPs.

## Figures and Tables

**Figure 1 ijms-21-06767-f001:**
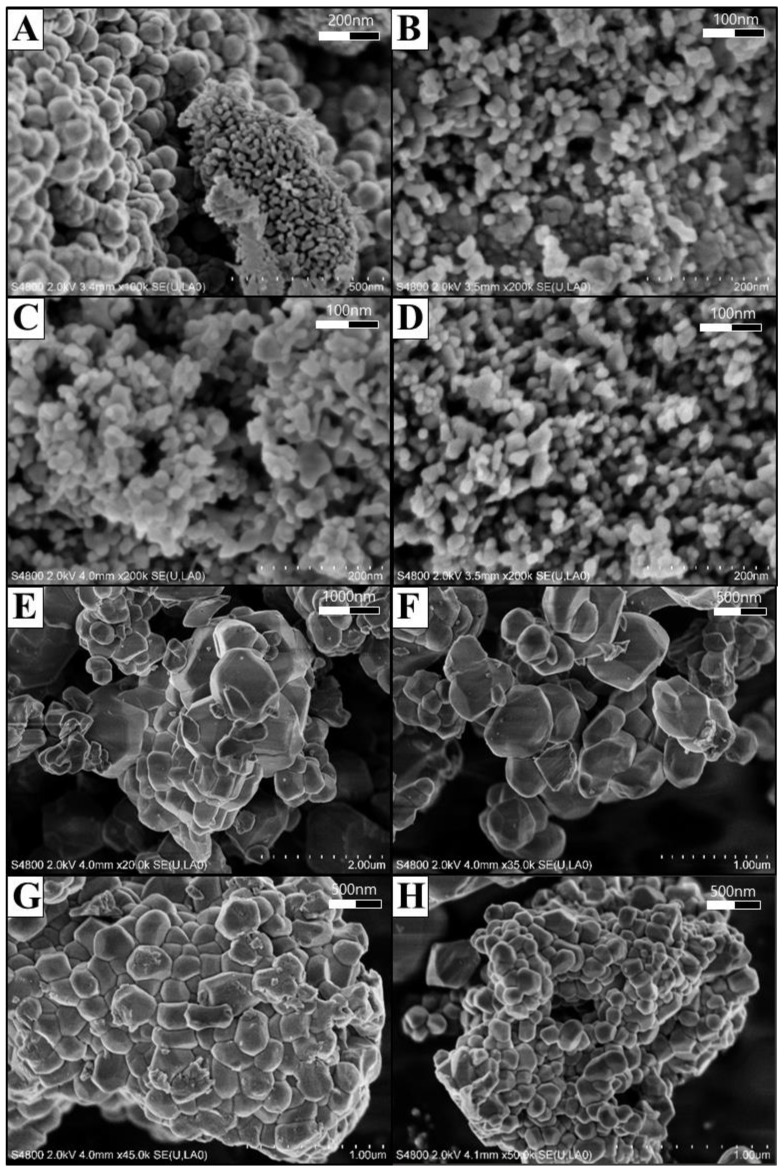
Field-emission scanning electron microscopy (SEM) images of the two different sizes of Co_3_O_4_ nanoparticles (NPs). (**A**) 100,000× and (**B**–**D**) 200,000× for type A Co-NPs; (**E**) 20,000×, (**F**) 35,000×, (**G**) 45,000×, and (**H**) 50,000× for type B Co-NPs.

**Figure 2 ijms-21-06767-f002:**
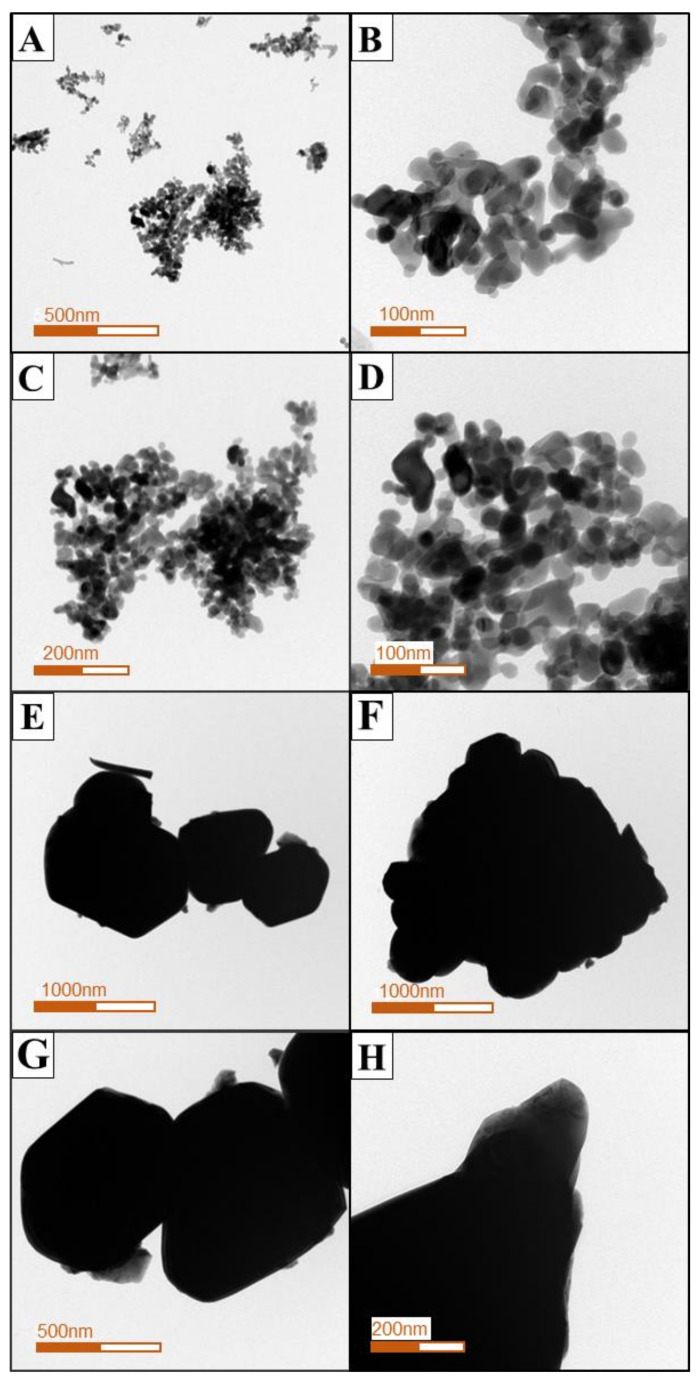
Transmission electron microscopy (TEM) images of the two different sizes of Co_3_O_4_ nanoparticles. (**A**–**D**) for type A Co-NPs; (**E**–**H**) for type B Co-NPs.

**Figure 3 ijms-21-06767-f003:**
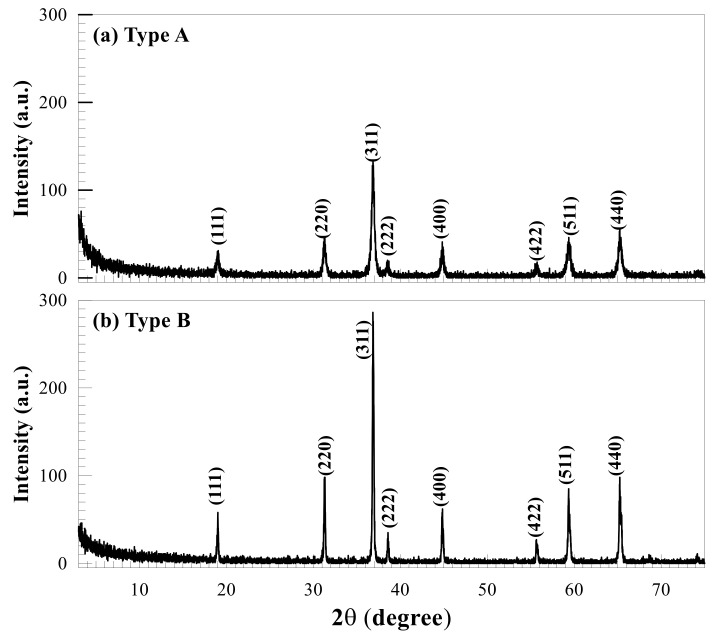
X-ray diffraction patterns of the two types of Co-NPs of different sizes. (**a**) for type A Co-NPs; (**b**) for type B Co-NPs.

**Figure 4 ijms-21-06767-f004:**
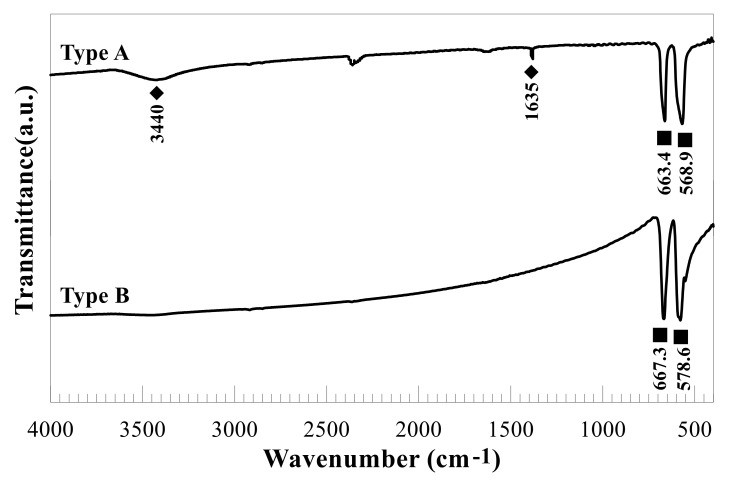
Fourier-transform infrared spectroscopy (FT-IR) spectra for the two types of Co-NPs of different sizes.

**Figure 5 ijms-21-06767-f005:**
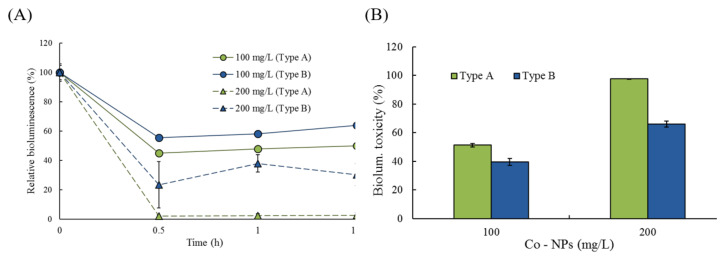
Dose–response (percentage of relative bioluminescence) curves of the effects of the two different sizes of Co-NPs on the bioluminescence activities of RB1436 during the incubation period: (**A**) relative activity changes during incubation periods, (**B**) comparisons of the toxicity based on the average of two measurements (1.0 and 1.5 h). Values are given as the mean ± SD (standard deviation) of triplicate samples.

**Figure 6 ijms-21-06767-f006:**
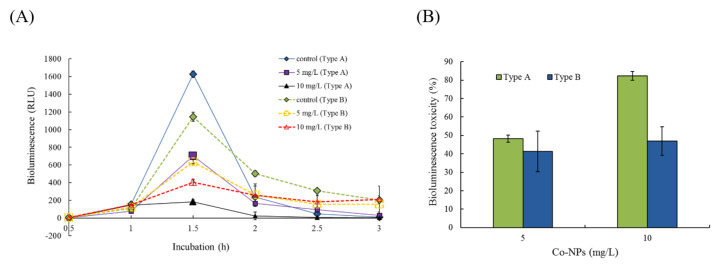
Comparisons of the effects of the bioluminescence activity of bioreporter strain KG1206 in the presence of the two different sizes of Co-NPs with inducer *o*-chlorotoluene (final concentration 1 mM): (**A**) changes in bioluminescence activity by incubation time, (**B**) comparison of the toxic effects on the bioluminescence activity.

**Figure 7 ijms-21-06767-f007:**
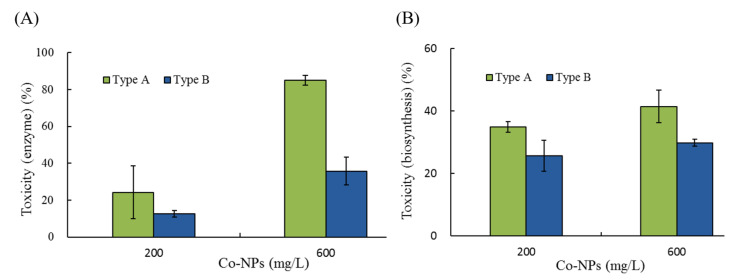
Comparisons of the effects of the two different sizes of Co-NPs on the enzymatic activity and biosynthesis of β–galactosidase: (**A**) toxicity on enzymatic activity, (**B**) toxicity on biosynthetic activity.

**Figure 8 ijms-21-06767-f008:**
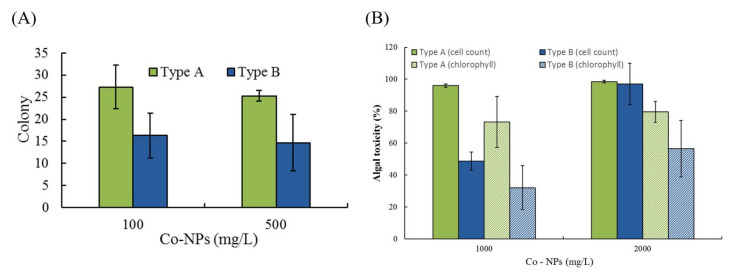
Comparisons of the toxic effects of the different sizes of Co-NPs: (**A**) on the colonies formed by reverse mutation, (**B**) on the two endpoints (cell count and chlorophyll content) of algal growth.

**Figure 9 ijms-21-06767-f009:**
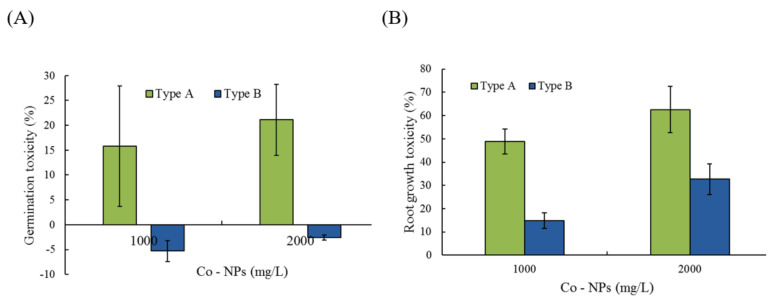
Comparisons of the toxic effects of the different sizes of Co-NPs on plant activity: (**A**) toxicity of seed germination, (**B**) toxicity of root growth of *Lactuca.* Comparisons were made based on the relative activity of the control set: no toxicity and the without Co-NP amended condition (soluble cobalt metal concentrations: 64 ± 14 μg/L and 439 ± 135 μg/L for the small particles (type A) and 5 ± 0.6 μg/L and 8 ± 1 μg/L for the large particles (type B) with exposures of 1000 mg/L and 2000 mg/L of Co-NP, respectively).

**Table 1 ijms-21-06767-t001:** The characteristics of the Co-NPs for the experiments reported by the manufacturer (MFR) and as particles determined by several methods.

Type	Size of Particle or Crystallite (nm)				Surface Area (m^2^/g)	
	MFR ^a^	XRD ^b^	SEM/TEM ^c^	SA ^d^	MFR ^a^	BET ^e^
A	10–30	26	10–30 (> 50)	25.3	40–70	38.94
B	50–80	69	80–150 (> 200, 500–1000)	1062	10	0.929

^a^ manufacturer’s value; ^b^ calculated by XRD (Scherer); ^c^ observation from SEM and TEM; ^d^ calculated by surface area measurement; ^e^ measured by the BET method.

**Table 2 ijms-21-06767-t002:** Summary of the average toxicities of eight biological activities in the presence of Co-NPs.

Conditions	Size 1 (type A)		Size 2 (type B)	
	low conc. ^b^	high conc.	low conc.	high conc.
Average toxicity (%) (except for shoot growth) ^a^	43 ± 7.0	63 ± 3.0	19 ± 5.5	32 ± 5.7
	53 ± 5.0		25 ± 5.6	

^a^ from the nine biological activities, shoot growth, which showed stimulation activity, was not counted for average toxicity; ^b^ conc. (concentration).

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
