# Peer review of "Comparative Effects of Particle Sizes of Cobalt Nanoparticles to Nine Biological Activities"

_ijms, 2020, doi:10.3390/ijms21186767_

Round 1
Reviewer 1 Report
Dear authors.
Photos from an electron microscope show the size of nanoparticles, but you can see that the nanoparticles are aggregated. Have you studied in which state nanoparticles are in liquid: as individual nanoparticles or as aggregates?
The article claims to be a systematic analysis of the influence of cobalt nanoparticles on different taxa of living organisms. Paper is fine, but I think that discussion section also can be improved. In particular, the discussion could be expanded by comparing different taxa reactions to the same nanoparticles.
DOI: 10.1016/j.plaphy.2016.04.024;
DOI: 10.1016/j.jhazmat.2020.123532;
DOI: 10.1155/2018/5263814;
DOI: 10.1016/j.jmst.2020.05.030;
DOI: 10.1016/j.envres.2020.109513;
DOI: 10.1016/j.jhazmat.2020.123542;
DOI: 10.1186/s13071-019-3528-2;
DOI: 10.2147/IJN.S257711;
DOI: 10.1080/17458080.2014.895061
I hope you will find useful ideas in these articles.
Author Response
Reviewer#1
- Photos from an electron microscope show the size of nanoparticles, but you can see that the nanoparticles are aggregated. Have you studied in which state nanoparticles are in liquid: as individual nanoparticles or as aggregates?
(Answer) All characterizations were performed using the particle states of NPs.
- The article claims to be a systematic analysis of the influence of cobalt nanoparticles on different taxa of living organisms. Paper is fine, but I think that discussion section also can be improved. In particular, the discussion could be expanded by comparing different taxa reactions to the same nanoparticles.
(Answer) Thank you for your valuable articles. Authors used some of these articles (five out of nine suggested ones) for the revision of this manuscript. All revised ones were marked with green color.
Reviewer 2 Report
Major comments:
- In the characterization of Co oxide nanoparticles, the morphologies of NPs were uniform and aggregation with different sizes, and the scale bars were not clearly in the Fig. 2, I suggest that the Fig. 2 need to remodify. On the other hand, the molecules and charge on the surface of particle did not well-define. It is not easy to understand and identify the main issue of NPs to lead the biological toxicities. In addition, why the type A of NPs have two more peaks (1635 and 3440 cm-1) that compare with type B in the FT-IR spectrum? and what the signal at 2400 cm-1 of type A NPs in the spectrum?
- Why used 100 and 200 nm NPs to evaluate the biological toxicities in these studies, the author did not describe clearly in the manuscript, I suggest that the author need to point the rationales in the manuscript.
- In the section of activities in bacterial system, the author mentioned that Fig. 5B was calculated after 1 and 1.5 h of exposure, but it was not pointed with different exposure time in Fig. 5B. In addition, the figure lacked of control condition.
- Following the Q2, why the author used different concentration between RB1436 (100 and 200 mg/L) and KG1206 (5 and 10 mg/L)? In the KG1206, 10-30 nm Co-NPs is same with type A? and it was lack of results of 50-80 nm Co-NPs in KG1206. The design of experiments was confused and missed some conditions in this part. Why the bioluminescence was increase at exposure time of 1.5 h with different conditions in the study of KG1206?
- I suggest that all of the illustrations need to presented as the mean ± standard error and compared between groups with statistical analysis, such as student’s t test or one-way ANOVA. For example, the author mention that the Co-NPs have effect on gene mutation in different condition with 2000 mg/L (Fig. 8B), if the author apply the statistical analysis, it would be not showing the significant difference.
Author Response
Reviewer#2: Major comments:
1. In the characterization of Co oxide nanoparticles, the morphologies of NPs were uniform and aggregation with different sizes, and the scale bars were not clearly in the Fig. 2, I suggest that the Fig. 2 need to remodify.
(Answer) We revised the scale bar in Figure 1 more clearly to avoid confusion in reviewers.
2. On the other hand, the molecules and charge on the surface of particle did not well-define.
(Answer) Unfortunately, we did not conduct the characterization of the surface properties of NPs such as the surface charge in this study. So we had no data for the description of the molecules and charge on the surface of NPs.
3. It is not easy to understand and identify the main issue of NPs to lead the biological toxicities.
(Answer) Yes, it is not easy to identify the exact causes of the toxicity. As many researchers discussed, the precise toxicity mechanisms of NPs remain largely unknown and complex. Detailed researches have to be continuously performed.
4. In addition, why the type A of NPs have two more peaks (1635 and 3440 cm-1) that compare with type B in the FT-IR spectrum? and what the signal at 2400 cm-1 of type A NPs in the spectrum?
(Answer) Answers for this question described in (Line 114-121).
(Additional Comments) The small sharp peak at ~1385 cm -1 is an artifact from the KBr pellet, and peak around 2400 cm-1 is from atmospheric CO2. Since these weak peaks are background noises originating from the measurement environment, the related information has been deleted from the text and figures.
5. Why used 100 and 200 nm NPs to evaluate the biological toxicities in these studies, the author did not describe clearly in the manuscript, I suggest that the author need to point the rationales in the manuscript.
(Answer) As mentioned in Materials and methods section and Table 1, we bought commercially available NPs, which have reasonable size differences. Those two sizes are 10-30 and 50-80 nm (not 100 and 200 nm) and clearly rationalized in Table 1.
6. In the section of activities in bacterial system, the author mentioned that Fig. 5B was calculated after 1 and 1.5 h of exposure, but it was not pointed with different exposure time in Fig. 5B. In addition, the figure lacked of control condition.
(Answer) The mean bioluminescence activity measured at 1 and 1.5 h during the incubation periods were used for the comparison of the toxicity. This protocol has been developed and used in our laboratory. All values were presented as relative bioluminescence of each control (Fig 5A). Based on the results of Fig. 5A, average values at 1 and 1.5 h were used for the toxicity evaluation.
7. Following the Q2, why the author used different concentration between RB1436 (100 and 200 mg/L) and KG1206 (5 and 10 mg/L)? In the KG1206, 10-30 nm Co-NPs is same with type A? and it was lack of results of 50-80 nm Co-NPs in KG1206. The design of experiments was confused and missed some conditions in this part. Why the bioluminescence was increase at exposure time of 1.5 h with different conditions in the study of KG1206?
(Answer) All bioassays performed at two concentrations, which was determined based on the preliminary experiment. Bioluminescence is produced during the growth periods for RB 1436, while in the presence of inducer for KG 1206. Same concentration of inducer were amended for all sets of KG 1206. Differences of maximum bioluminescence activity rely on the toxicity of exposed NPs. The decrease of bioluminescence production after maximum activity was due to the used-up of inducer amended.
8. I suggest that all of the illustrations need to presented as the mean ± standard error and compared between groups with statistical analysis, such as student’s t test or one-way ANOVA. For example, the author mention that the Co-NPs have effect on gene mutation in different condition with 2000 mg/L (Fig. 8B), if the author apply the statistical analysis, it would be not showing the significant difference
(Answer) All comparisons were made statistically analysis using student t-test. Throughout the experiment, the statistical differences sometimes depend on the concentration. “In particular, significant statistical differences in toxicity were observed between the different sizes for both observed endpoints at the low concentration (1000 mg/L) tested, showing nearly 2.1-times higher algal growth toxicity of small NPs compared to that of large NPs (p = 0.0117).” Fig 8A and 8B are the results of gene mutation and algal growth, respectively. As reviewer mentioned, the statistical difference was observed at low concentration tested (1000mg/L), but not at high concentration on algal growth because of great toxic for both particle sizes.
Reviewer 3 Report
Referee Report
Manuscript number: ijms-907734
Title: Comparative Effects of Particle Sizes of Cobalt Nanoparticles to Nine Biological Activities
By Kong et al
Submitted to IJMS
Comments:
From the title, this manuscript is to investigate the size effect of Co NPs regarding nine biological activities. I have the following concerns in this work:
- There are only two sizes of NPs investigated in this study. More sizes of particle should be investigated as this work is related to the size effect.
- From the title, why did the authors emphasize to study “nine” biological activities? Why it is not other numbers and why such number (nine) is important in nano-biology.
- Biology is a board field and the authors may consider other terms regarding the application of the Co NPs. For example, the size effect in functional imaging, drug delivery, therapy and so on. I noted that the authors focused on the cytotoxicitiy of the NPs but they did not mention it in the title of the manuscript.
- Materials and Methods section should be in front of the Results section.
- Figure 1(E-H): I can see the sizes of particles are in the order of micrometer but not nanometer.
- Figures 2D and 2H: The scale bar can be located in the top-right-corner. In addition, there are “marks” on the left of the scale bar in Figure 2A, B, E, F and G.
- Table 1: Please check if the unit of the surface area is (m^2/g)?
- Figures 5 and 6: Please add error bars to all data points.
Author Response
Reviewer#3
Comments:
From the title, this manuscript is to investigate the size effect of Co NPs regarding nine biological activities. I have the following concerns in this work:
1. There are only two sizes of NPs investigated in this study. More sizes of particle should be investigated as this work is related to the size effect.
(Answer) If we are able to test more sizes of particles, more scientific results hopefully could be obtained. But it is nearly not feasible to find commercially available ones at this moment. In addition, all sets were performed in triplicate, so testing with more than two particle sizes is nearly impossible to handle to get a proper results. The main purpose of this study was to compare the potential effects of particle sizes and showed the high toxicity of small particle sizes. In the future, various sizes of NPs can be tested based on this investigation.
2. From the title, why did the authors emphasize to study “nine” biological activities? Why it is not other numbers and why such number (nine) is important in nano-biology.
(Answer) As reviewer commented, the number “nine” is not important, but authors tried to inform the numbers of biological systems tested in this study.
3. Biology is a board field and the authors may consider other terms regarding the application of the Co NPs. For example, the size effect in functional imaging, drug delivery, therapy and so on. I noted that the authors focused on the cytotoxicitiy of the NPs but they did not mention it in the title of the manuscript.
(Answer) Authors thought about the proper title prior to present this article. This title was named based on the published research articles, which are similar to our topics. But, if reviewer suggested any proper one, we can consider for the change of title.
4. Materials and Methods section should be in front of the Results section.
(Answer) Unlike many other journal, IJMS required the section “Material and Methods” after discussion section.
5. Figure 1(E-H): I can see the sizes of particles are in the order of micrometer but not nanometer.
(Answer) We revised the scale bar in Figure 1 more clearly to avoid confusion in reviewers.
6. Figures 2D and 2H: The scale bar can be located in the top-right-corner. In addition, there are “marks” on the left of the scale bar in Figure 2A, B, E, F and G.
(Answer) We revised the scale bar in Figure 2 more clearly to avoid confusion in reviewers.
7. Table 1: Please check if the unit of the surface area is (m^2/g)?
(Answer) The unit of surface area in Table 1, which are measured in the study and provided by manufacturer, is m2/g. We also revised the unit of m2g-1 in (line 117) to m2/g to ensure unit uniformity.
8. Figures 5 and 6: Please add error bars to all data points.
(Answer) All data points were expressed with error bars (std) to give statistical information. But, some (e.g., few data points of figure 5 and 6) error bars were too small to see at figures.
Round 2
Reviewer 2 Report
Based on the author’s response, the author mention that the comparisons were made statistically analysis using student t-test, and the p= 0.0117, are the same of the all comparison in this manuscript? I still suggest that the author need to mark the variation of all comparisons and point out the p values in the figure captions.Author Response
Comments: Based on the author’s response, the author mention that the comparisons were made statistically analysis using student t-test, and the p= 0.0117, are the same of the all comparison in this manuscript? I still suggest that the author need to mark the variation of all comparisons and point out the p values in the figure captions.
(Answer) Thank you for your valuable comments.
Authors have tried to detail information about p-values of each condition in the text (e.g., line 151 p = 0.0001-0.0070; line 165 p = 0.4874; line 167 p = 0.0022; line 194 p < 0.0191; line 208 p = 0.0194; line 235 p = 0.0117; line 247 p = 0.0045; line 257 p = 0.0017, p = 0.0012). If reviewer doesn’t mind, authors would like to keep the p-values as described in the text. This is because it shows various values depending on the tested concentration, therefore, it is difficult to point out the p values in the figure captions.
Reviewer 3 Report
I accept the answers from the authors based on my comments and the corresponding modifications in the manuscript. I understand the limitation explained by the authors in the experimentation.
Author Response
Comments: I accept the answers from the authors based on my comments and the corresponding modifications in the manuscript. I understand the limitation explained by the authors in the experimentation.
(Answer) Thank you for your understanding and efforts.